# The Future and Sustainability of Carpooling Practices. An Identification of Research Challenges

**Anne Aguiléra * and Eléonore Pigalle**

Laboratoire Ville Mobilité Transport (LVMT), Université Gustave Eiffel and ENPC, 77454 Marne-la-Vallée, France; eleonore.pigalle@univ-eiffel.fr

* Correspondence: anne.aguilera@univ-eiffel.fr

**Abstract:** This article proposes several new research directions regarding the future and sustainability of carpooling practices. The reflection is based on a systematic review of the literature (2010–2021) and a consideration of some of the main recent changes in carpooling, such as carpooling platforms and apps, and changes in lifestyles that may affect carpooling practices, such as telework. Carpooling is defined here as the sharing of short- or long-distance car rides between people who are not members of the same household, for a trip (or part of a trip) already scheduled by the driver, free of charge or expense sharing. After a description of the corpus selection method used, followed by a brief review of existing literature, we propose three new avenues for research. The first avenue is a call for greater consideration of forms of transport, other than carpooling, to work (or to the place of study), which remain poorly studied. The second avenue proposes the investigation of the role that the current digitalisation of carpooling, namely online carpooling platforms and their integration into mobility platforms, and also the use of social media by carpoolers, could play in the coming years. The third avenue argues that the consequences of the rise of collaborative consumption and the current pandemic, namely teleworking practices and the perception of shared mobility, could have an effect on carpooling, which merits specific investigation.

**Keywords:** carpooling; mobility; collaborative consumption; apps; social media; MaaS; teleworking; sustainability; pandemic

## 1. Introduction

In its broadest definition, carpooling refers to shared rides by car between people with similar origin–destination pairings [1–4]. It is different from ridehailing (and ridesplitting, i.e., shared ridehailing), in which people pay for a ride from a professional or part-time driver through an app [5]. Indeed, in carpooling, the driver is not a professional but a private individual who agrees to share his journey (or part of it) for free or for a contribution to the travel costs. Moreover, carpooling must not be confused with carsharing, a vehicle access scheme in which people obtain access to a car they do not own for a period of time [6].

Carpooling includes different categories and is the subject of various typologies in the literature [7]. They generally take into account trip purpose (work or non-work), methods of matching between drivers and passengers (IT-based or casual) and if the driver and the passenger(s) belong to the same household or not [2,4]

In this article, the term is used in a more restricted sense since it excludes household carpools and family carpools, which currently account for the majority of carpooling practices [8–10]. Therefore, in the remainder of the text, carpooling refers to the sharing of car rides between people who are not members of the same household, for a trip (or a part of a trip) already scheduled by the driver and not for the purposes of profit (although costs

of travel may be shared). Both carpooling for work and non-work purposes as well as IT-based and casual carpooling are considered.

Defined in this way, carpooling has been a matter of interest for researchers and public authorities for decades [11–13]. The main reason is that carpooling is often considered as a low-cost option to decrease car ownership and solo driving along with their major environmental impacts, such as congestion and air pollution [4,14]. Although less documented, social benefits, such as improved accessibility for low-income people and social cohesion, are pointed out by some authors [15–17]. However, the literature has revealed the existence of many barriers to carpooling practices [18] and highlighted its uncertain impacts on travel behaviour, especially car ownership and use [13,19–23]. Nonetheless, the recent rise of IT-based carpooling, i.e., based on platforms and apps, such as Blablacar [24], has rekindled interest in the subject and, especially, given hope that the downward trend in carpooling observed in Europe and North America [2,25] would be reversed.

Without pretention of exhaustiveness, this article proposes several new research directions regarding the future and sustainability of carpooling practices. The reflection is based, on the one hand, on a systematic review of the literature (2010–2021) in social sciences using Google Scholar and a list of relevant keywords, and, on the other hand, on a consideration of some of the main recent changes in carpooling and in lifestyles that may affect carpooling practices.

After a description of the corpus selection method used (Section 2) followed by a brief review of existing literature (Section 3), we propose three new avenues for research. The first avenue is a call for greater consideration of non-work carpooling, i.e. forms of carpooling other than carpooling to work (or to the place of study), which remain poorly studied (Section 4). The second avenue proposes the investigation of the role that the current digitalisation of carpooling, namely the use of digital tools (online carpooling platforms, mobility platforms but also social media), by carpoolers could play in the coming years (Section 5). The third avenue argues that the consequences of the rise of collaborative consumption and the current pandemic, namely teleworking practices and the perception of shared mobility, could have an effect on carpooling, which merits specific investigation in the near future (Section 6). Finally, the conclusion summarises the main findings and discusses some policy implications of these new research directions.

## 2. Corpus Selection

Relevant academic literature was identified with the Google Scholar search engine with the keywords "carpool", "carpooling", "carpooler", "carpoolers", "ridesharing", "ride-sharing" or "rideshare" in the title. Indeed, there is sometimes a confusion between carpooling, ridesharing and ridesourcing services in the literature. The search period was 2010 to 15 August 2021. At the end of this first stage, 1671 documents were identified.

Based on their title or, if not sufficient, their abstract, we excluded the numerous papers focusing on ridesourcing; those focusing exclusively on household carpools or family carpools, or (more often); those for which it was not possible to differentiate these two categories of carpooling from non-household carpooling (for instance in the numerous papers about HOV lanes). Furthermore, we also excluded purely technology-oriented papers, such as the (very numerous) papers focusing on the best algorithms to match drivers and passengers. Based on our expertise in the field, we also added a few influential papers published before 2010, such as [3] and two relevant recent papers published only in French. Finally, we also excluded the reports and the papers not published in an academic journal or a book, with four exceptions regarding their interest for our analysis: two conference proceedings papers [22,26] and two working papers [27,28]. Our final corpus encompassed 97 papers, mainly published in academic journals.

We then proceeded to perform a comprehensive analysis of the content of these 97 selected papers. Firstly, we identified the main topics and results already addressed by the literature. They are briefly described in Section 2. This initial analysis was then used to identify research gaps and challenges about the future and sustainability of carpooling

practices. They constitute the core of this paper and are successively discussed in Sections 3 to 5, as announced in the introduction.

## 3. Brief Literature Review

As shown by the size of our corpus and the two most recent meta-analyses on the topic [7,18], carpooling has long been a topic of numerous academic studies during the last decade. Our analysis of the main research topics underlines that many of the selected papers have investigated the profile of carpoolers [29–32] —or, more often, people expressing an intention to carpool [1,33–36]—compared to non-carpoolers, and their main motivations to practice carpooling [13,37]. Findings revealed that psychological dimensions, such as attitude towards carpooling, enjoyment of being sociable, trust in other people, environmentalist identity and the role of the family circle and peer group, are more influential than classical socio-demographic, economic and spatial variables [38–42]. Cultural elements, such as car culture [43], also play a role in explaining carpooling behaviour, but they were far less investigated [44]. In addition, the influence of several classical variables on carpooling intention or practice, such as gender and the characteristics of the place of residence (density, urban *versus* rural), is still disputed as shown by the two most recent meta-analyses on the topic [7,18].Finally, motivations to carpool are both individualistic, such as flexibility (compared to public transport, for instance), time and cost benefits or fear of driving and limited parking places [45], and collectivistic such as environmental sustainability, sociability and the desire to help other people [46,47].

Numerous studies have also sought to assess the impact of different types of carpooling services (in terms of quality of service such as flexibility, price, etc.) and incentives introduced by governments, universities, employers or transit agencies: dedicated lanes on highways, reserved parking places or tradable credit schemes [48–54]. Research has also intensively looked at the numerous obstacles to carpooling, such as difficulties in finding a passenger or a driver and in matching schedules, but also fears or discomfort of sharing a trip with a stranger [18,55,56].

Finally, the literature has questioned the sustainability of carpooling practices, mainly from an environmental point of view and with contrasting results [20–22,57–59]. If some authors conclude that carpooling facilitates a reduction in car use or car ownership and hence $CO_2$ emissions [19,23,60–63] , others obtain less favorable or even opposite results: in particular, they highlight that carpooling can be in competition with public transport rather than with the car [3,13,64]. For example, [13] highlighted that 70% of carpoolers in the San Francisco region were former users of public transport, and only just over 10% were former car users. On the contrary, [58] found important energy savings in San Francisco. Finally, some authors question the factors favouring the development of carpooling to feed public transport [65,66]. The impacts of carpooling practices in terms of the reduction in household car ownership seem more consensual and positive [19,23].

By comparison, the social dimensions of carpooling remain less documented [17], although several studies underline that social motivations are as important as economic motivations among carpoolers [12] and that carpooling practices can be linked to social relations within a neighbourhood, although the links between the two remain complex [15] and can contribute to the reduction of transport exclusion [67]. Finally, safety aspects of carpooling (compared to solo driving) such as speed reduction and lower use of the smartphone while driving have been little investigated [68].

## 4. The Under Consideration of Carpooling for Non-Work Purposes

Although very rich, the literature has primarily focused heavily on regular and short-distance commuting trips, i.e., carpooling between home and the workplace (or the place of study: school or university); this can, for instance, be observed in the recent meta-analysis by [18]. By comparison, carpooling for non-work trips was less considered, although the success of the Blablacar platform has recently turned the spotlight on long-distance carpooling, which primarily concerns non-work purposes [4,69,70].

This interest in carpooling to work is legitimate: commuting trips are on average longer and more frequently made by car than other trips. They are therefore responsible for high levels of pollution and considerable negative impacts on human health [71]. They are also highly concentrated in time, causing daily traffic jams in the world's big cities that reduce the economic efficiency of the urban concentration model [72]. As a result, the development of carpooling is generally considered to be an instrument for decreasing congestion and decarbonising travel behaviour [73]. Other reasons for the interest in carpooling are the spatial and temporal regularity of commuting trips (which may facilitate the organization of shared trips), their costs for the commuters and the fact that companies are good places to form driver–passenger pairs and for employers to provide incentives.

By comparison, much less is known about non-work carpooling for both short- and long-distance trips [12]. However, both the environmental and social issues associated with these trips are significant. In the US, for example, daily shopping trips and social and recreational trips are more numerous than commuting trips, and they are primarily made by car [74]. In social terms, the development of everyday and long-distance carpooling has the potential to improve access to mobility for people with health or economic problems, or people living in areas with poor public transport provision [75,76]. Sharing rides associated with childrens' school and extracurricular activities is also a way to ease the pressure on complex family schedules [47]. Therefore, future research should consider carpooling for non-work purposes with more attention. We propose the following three main research directions.

The first concerns the measurement of non-work carpooling practices. In particular, household travel surveys have rarely been used for that purpose and few figures are available on non-work carpooling practices in terms of frequency or modal share. Some recent studies in France suggest that non-work carpooling is widespread among the population, although infrequent [12,77]. They also show that most carpoolers practice both work and non-work carpooling, something that has so far attracted very little attention in the literature, which tends to consider carpooling practices in isolation.

The second research direction concerns the profile and motivations of non-work carpoolers, both passengers and drivers. Even if comparisons are still scarce, there seems to be a significant difference in profiles and motivations between people carpooling for work and for non-work purposes. Based on a survey among French drivers, [12] highlighted that young people were over represented among people sharing car rides for short-distance work trips, while people who shared rides for shopping purposes were in average older. In addition, women were over represented among people who used carpooling for children-related trips. In a recent study among French adults, [77] concluded that people carpooling for short-distance, non-work trips formed an older, less urban and lower earning group that people carpooling to work or for long-distance trips. In a study among Blablacar users in France (long-distance carpooling), [24] demonstrated that, compared to the French population, long-distance carpoolers had the same average income level but were more educated (most of the respondents had a university diploma), much younger and more frequently resided in rural municipalities. In addition, their study revealed that most shared trips were for leisure purposes. Moreover, mutual help seems to be an important motivation to share non-work related car trips, especially in areas with poor public transport provision or in communities marked by existing forms of solidarity [78]. For long- and short-distance, work-related carpooling, although there was a social component, the financial aspects seemed more influential. However, in another recent study among Blablacar users, [79] demonstrated how social value was important to explain the continuance of use of this service. Undoubtedly, these aspects merit further investigation.

Thirdly, more research is needed on the environmental and social impacts of non-work carpooling. Recent studies on long-distance carpooling, which concerns primarily non-work purposes, show competition between carpooling and the train [80,81]. With regard to daily, non-work trips, the conclusions are less clear and research is still lacking [12,77]. In particular, we lack studies using longitudinal or biographical data [82], which

will make it possible to evaluate if and how the adoption of carpooling is associated with gradual changes in modal choices and car ownership at the individual and household levels in the short-, medium- and long-terms [83]. Finally, as for carpooling to the workplace or the place of study, social issues of non-work carpooling are poorly documented. Even if some studies have analysed the social motivations of carpoolers, such as conviviality and enjoyability [8,35] a comprehensive analysis of the social dimensions and impacts of non-work carpooling, especially regarding social integration and cohesion [84], accessibility [85] and also discrimination issues in the choice of a driver or a passenger [86], is still missing [17].

## 5. The Impacts of the Digitalisation of Carpooling

This second section looks at the new research avenues opened up by what we propose to call the digitalisation of carpooling, i.e., the use of digital tools by carpoolers. For us, this notion covers but is not limited to, carpooling platforms and mobility platforms that include carpooling apps, although they are a central feature. The digitalisation of carpooling also, in our view, operates *via* social media, through processes that have so far received little scholarly attention. One such process is the creation of carpoolers groups, and another is the contribution of social media to the spread of new social norms around a practice in which the importance of subjective factors has largely been demonstrated [40].

### 5.1. Carpooling Platforms and Apps

Many mobility services are now available on smartphones [87]. Some can be used to obtain real-time information on transport systems (roads and public transport) and recommendations on the best choice of routes and transport modes. Others provide access to new forms of shared trips (ridesourcing services, carpooling) or vehicles (carsharing, self-service bicycles, etc.).

In this new landscape, carpooling is attracting new actors [88–90]. Whether it functions as a planned or dynamic (i.e., real-time) service, the (many) platforms and apps now available usually focus on a specific segment of mobility: commuting, long distance or, more rarely, everyday non-work trips. Drivers and passengers are matched either by means of a standard search engine (where they enter their trip criteria, times, etc.), or automatically. In other kinds of application, an algorithm does the job of determining the (best) driver–passenger matches [8]. In general, these online applications also include a tool for sharing travel costs, plus a commission payable to the platform if it is operated by a private entity [91].

In theory, these applications offer numerous advantages for both drivers and passengers [92]. They include multiple options to improve the chances of matching supply and demand in both quantity and quality [93], including automatic ridematching processes [94]. They also create possibilities for new carpooling services such as "last-minute" pairing (dynamic carpooling) [95,96] and multi-hop carpooling [97]. The smartphone also facilitates coordination between carpoolers, for example if one of the parties is running late. In addition, platforms and apps offer transparent access to the prices proposed by drivers (hence the possibility of negotiating them downwards), as well as a system of online payment that helps to prevent both last-minute cancellations and the need for exchanges of money between individuals, which is unpopular in casual carpooling. Finally, systems for rating drivers and passengers serve to increase user trust [98,99], a very significant obstacle in casual carsharing [99–101]. Moreover, some applications undertake the responsibility of finding an alternative (such as a taxi) in the event of a last-minute cancellation by the driver. However, most of these apps do not currently attract much custom and are therefore not viable, often supported by public subsidies in experiments that generally do not last long [102].

While there is a huge literature on the technical aspects, i.e., improvements in the matching algorithms and in the design of the apps, the analysis of our corpus suggests three other research challenges.

The first concerns the profile of the carpoolers using apps and platforms to find a driver or passengers, and to make comparisons with casual carpoolers. In particular, it would be interesting to determine if IT-based carpoolers are former casual carpoolers or new carpoolers, and, more broadly, if the adoption process of IT-based and casual carpooling are similar or not.

The second research challenge focuses on the motivations for using IT-based carpooling and on the obstacles that remain. Recently, [8] highlighted how some carpooling apps based on automatic matching of drivers and passengers amplified psychological barriers to carpooling. Several years ago, [103,104] highlighted some limitations of the Internet for carpooling to work. These findings are obviously not definitive, and merit further investigation. They are recent and based on a very limited number of case studies. Some platforms and applications could gradually improve, become better known and then become more used. Indeed, it is worth remembering that Blablacar took around 10 years to achieve the public success it now enjoys.

The third research challenge concerns the impacts of IT-based carpooling on travel behaviour and especially modal share and car ownership, compared with casual carpooling. An important issue is to determine whether IT-based carpooling is capable of making carpooling more frequent and of convincing car users to abandon solo driving, especially in low-density areas [105].

*5.2. The Inclusion of Carpooling into Mobility Platforms*

The use of carpooling platforms and apps and their impact on modal share and car ownership will also have to be analysed within a new, broader ecosystem of platforms that include a growing number of mobility applications. The market for platforms that include various mobility services is in fact growing fast, simultaneously attracting actors in the digital and transport sectors, and the public authorities, which see them as a tool for decarbonising travel [106,107]. These platforms are often referred to by the generic term MaaS (mobility-as-a-service). The most sophisticated versions propose monthly or annual subscriptions to mobility services, which in general offer unlimited access to public transport and a limited access to ridehailing and carpooling platforms and apps [105] and micro-mobility services (such as shared e-scooters, shared bikes, etc.). Hence, these mobility platforms offer possibilities for a more sustainable business model for carpooling, especially short distance carpooling, which is currently one of its weakest aspects. In particular, carpooling could become a subscription service and/or a publicly subsidised solution, since digital technology can record evidence of carpooling and therefore reduce the risks of fraud. We propose two main research directions.

Firstly, more research is needed on the capacity of these mobility platforms to contribute to the development of IT-based carpooling in the areas where they operate in circumstances where, as we said before, large numbers of applications exist but most remain relatively unknown. Secondly, research is needed on the analysis of the environmental and social impacts of such evolution [108–111]. In particular, we need quantitative and qualitative studies on the profile, motivations and travel behaviour changes of MaaS platforms users, and more specifically on the place of IT-based carpooling for work and non-work purposes in their new travel behaviours.

*5.3. Carpooling and Social Media*

Social media now play an important role in the establishment and maintenance of relations between individuals in both the personal and professional spheres [112,113]. They also contribute to the formation of opinions and social norms, for example relating to privacy [114,115]. In the field of travel behaviour, these networks firstly provide researchers with substantial volumes of new data [116], including carpooling [39], and are

secondly helping to transform travel practices in ways that are only now beginning to be explored in the literature [117,118]. In particular, there are few studies dealing with the links between social media and (casual) carpooling [26,119]. In that respect, two prospects seem to us to be priorities in the coming years.

The first looks at the influence of social media on the formation and development of groups—in particular lasting groups—of carpoolers. Research has begun to show the influence of social networks, based on friendship circles or affinities around shared interests such as sport, culture, work, etc., on carpooling [28,78,84,120]. However, the influence of social networks has yet been little studied [26,119], or looked at from a theoretical perspective [121,122]. However, the advantage of this form of interaction, compared with carpooling platforms and apps, is that the members of these groups know each other more or less directly (i.e., through co-optation effects), and have already met (at least virtually). This richer mode of relationship not only helps to ease the problems of trust between carpoolers, but also makes it possible to adapt carpooling to common, and also very specific, mobility needs (e.g., carpooling for the childrens' school run). It should also be noted that these networks are highly responsive in the event of coordination problems such as an unexpected schedule change. Finally, the members communicate with each other for purposes other than carpooling and can therefore develop trust and relationships and interdependencies in other domains, potentially anchoring carpooling in broader—and therefore perhaps more lasting—relations of friendship or mutual support. In other words, social media could become an instrument for the lasting spread of casual carpooling, by supporting the formation of communities of carpoolers, some of whom maintain strong bonds in other contexts.

Moreover, and this is our second avenue of research, social media could help to change perceptions, and therefore social norms, around both IT-based and casual carpooling, by disseminating views about their advantages or, conversely, their downsides, on the Internet [123]. This factor seems to us particularly relevant insofar as the influence of private or occupational peer groups on the adoption of carpooling has been demonstrated in previous studies [38,77,117]. The task now is to determine whether social media could, in certain areas or for certain populations, fulfil similar or complementary functions for carpooling.

## 6. The Impacts of the Diffusion of Collaborative Consumption and of Health Crises

This third and final section relates to the current research field and how new lifestyles contribute to changing travel behaviour [124–126]. While the literature on these subjects is booming, especially regarding the current pandemic, our corpus indicates that it has largely neglected the question of carpooling with the exception of recent research on the impacts that shared autonomous vehicles could have on its development [127,128].

Yet, we believe that at least two recent or emerging lifestyles could have an impact on the practice of carpooling, both IT-based and casual. The first concerns the general rise of peer-to-peer sharing practices, the development of which could prove to be a direct or indirect benefit in the field of shared mobility. The second concerns recent heath crises with, on the one hand, the expansion of teleworking practices and their consequences on travel behavior [129,130] and, on the other hand, the changes the pandemic is having regarding modal choice [131].

### 6.1. Collaborative Consumption

Our first question concerns the capacity of carpooling to become one among a multiplicity of other collaborative consumption practices [132,133] through mechanisms of dissemination into mobility practices [134]. Considering carpooling not only as a mobility practice but also as a collaborative practice opens new research perspectives. In our view, two directions can be explored in that respect.

Firstly, the growing use of collaborative consumption platforms in other everyday domains could make it more natural to use it for purposes of short- or long-distance travel,

and also more attractive, as a result of successful prior experiences. In other words, IT-based carpooling could benefit from spillover effects. Recently, [135] showed the existence of close interconnections between AirBnB and the Uber and Lyft ridehailing services in several US tourist cities. In France, [77] highlighted that work, non-work and long-distance carpoolers were more likely to use collaborative consumption platforms than non-carpoolers. In Egypt and in China recent studies confirmed positive links between collaborative consumption and carpooling practices [136,137]. However, there has so far been little documentation on the possible complementarity effects of collaborative consumption practices in the field of mobility. They constitute an interesting direction for future investigations.

Secondly, involvement in the sharing economy has a strong symbolic dimension [138]. It reflects forms of adherence to alternative values relating to the organisation of the economy and social relations, values characterised by greater respect for individuals and the environment. The practice of carpooling is also very closely linked with subjective factors [7]. The question of the links between the representations associated with involvement in the sharing economy (in general) and the practice of carpooling therefore merits investigation. The recent findings by [77] with respect to carpooling in France are entirely consistent with this. The authors highlighted that, compared to non-carpoolers, carpoolers were more likely to share social and political values associated to collaborative consumption. Additionally, [139] recently identified three styles of collaborative consumption practices among carpoolers: communal collaborative consumption in which participants seek pro-social relationships, consumerist collaborative consumption where participants seek status and convenience and opportunistic collaborative consumption where participants seek monetary gain or personal benefits. Finally, [70] in a recent study of Blablacar users, highlighted that long-distance carpoolers used the values of the collaborative economy (such as the feeling of belonging to a community) to justify their practice.

These initial studies need to be followed by further research. Establishing links between carpooling and wider sharing practices between individuals seems to us to be an important and novel avenue of investigation. In particular, it could lead to a better understanding of how it is that carpoolers on average show greater environmental awareness than non-carpoolers, even though ecological motives at first sight seem very secondary in their decision to carpool. This research could also cast new light on the close connection between economic motives and motives of a more social nature (such as the enjoyment of company and mutual help) in the practice of carpooling, a connection that is characteristic of sharing economy practices in many domains [140]. More broadly, this type of analysis seems important to us for advancing our understanding of the multiple subjective dimensions of carpooling.

### 6.2. Teleworking

In recent months, the development of teleworking (and especially homeworking) has been spectacular. While the pandemic crisis has been the main driver, many experts think that teleworking, at home or in telecentres located close to the homeplace [141], has a fine future ahead of it, after decades of stagnation in many countries [124]. The reasons for this development are not only health related but also economic, since home-based telework is a way for some employers to save office space or relocate [142].

It is frankly difficult at present to predict what teleworking will look like tomorrow, what categories of employers and employees will be affected or indeed what forms it will take in companies (in particular, how many days a week) [143,144]. However, its impact on mobilities is already being examined very closely by academics and public authorities [145,146]. The latter generally hope that it will lead to a decline in trip frequency and transport congestion. However, well before the COVID-19 crisis, studies had highlighted the many rebound effects of teleworking on activity–travel patterns, especially non-work related trips, and on urban sprawl (which, it is now well known to stimulate car use) [146,147].

However, these findings need to be revisited, because tomorrow's telework will probably not be the same as yesterday's. In particular, it will probably be more regular (one to several days per week) and be extended to new occupational categories. The consequences for mobility practices, and particularly for the ownership and the use of the car, remain very uncertain for the moment [147–149]. An increase and a decline in the market share of the car seem equally plausible, and the direction of travel will depend both on the lifestyle changes that workers make (as a result of spending more time in their home communities), but also on whether or not the fear of infection associated with public transport persists. Impacts will also depend on future urban and transport policies that could use the teleworking argument to further restrict car use [150].

In these new circumstances that will probably lead to important reorganizations of activity–travel patterns, the future of carpooling merits exploration. Prior research (before the pandemic) highlighted that the adoption of home-based teleworking increased the frequency of non-work trips, especially in and around the municipality of residence, because of the increased time available through decreased commuting trips. There was, however, important differences between full- and part-day teleworkers [124,129,151]. A relationship with urban sprawl was also demonstrated by some studies, although the sense of causality remained unclear [124,152]. Moreover, impacts on modal choices were disputed and depended on the place of residence (urban, suburban or rural) [153]. However, to our knowledge, the relationship between teleworking (before the pandemic) and carpooling has not been much investigated in the past. Yet, we believe it is an interesting research topic for the future [27,154].

Several (contrasting) hypotheses can be formulated. Short- and long-distance carpooling to work could become more popular if commuting trips become less frequent and therefore easier to share (due to less organisational constraints), or because of the fear of taking public transport [155], which may prompt current users (teleworkers and non-teleworkers alike) to find new mobility solutions without bearing the cost of purchasing and maintaining a car. However, carpooling to work could also decline as a result of the continuing fear of infection in shared spaces or because of a fall in financial revenues when passengers no longer need to commute so regularly. Non-work trips, which are likely to rise, could also be impacted in ways that remain largely unknown. In particular, carpooling for non-work, but also work, purposes could increase as people spend more time and undertake more activities in their municipality of residence and are therefore likely to develop their local social networks, which could lead to the development of casual carpooling and reach not only environmental but also social targets. The very recent study by [154] in the Twin Cities area of Minneapolis–Saint Paul concludes that teleworkers are more likely to carpool for work trips than non-teleworkers. However, the empirical study was made before the pandemic.

Short-, medium- and long-term impacts of teleworking on carpooling are complex and still uncertain. They will undoubtedly vary across populations (according to professional status, household composition, psychological aspects, etc.) and territories (urban, suburban, rural), depending on local context such as urban form (density, diversity etc.) and transport systems, but also public policies, which are well-known drivers of travel behaviour and, especially, mode choice. In the years to come, these questions will become very important. In particular, research is needed to identify both new opportunities and new obstacles to the development of carpooling for work and also non-work purposes that the likely development of teleworking in many private and public organisations may generate. Findings will be particularly useful for public authorities and employers seeking to reduce car use.

### 6.3. Modal Choice after the Pandemic

Travel impacts of the pandemic remain highly uncertain and need further investigation regarding the demand for private versus shared transportation modes [156]. None-

theless, shared modes are threatened since they represent a potential risk of contamination. In that respect, public transport has received more attention than other shared transport modes. Huge ridership reductions were observed in many countries across the world, during but also after lockdown periods [157]. Studies are also needed on the impacts on the other shared modes of transport, such as carpooling. In particular, carpooling could be less impacted than public transport or it could even benefit from the situation since it could be considered less risky than public transport [32,158,159] due to fewer people or because carpooling apps could require sanitary guarantees such as a health pass. On the other hand, carpooling could be considered a risky transport mode compared to solo driving. In particular, the pandemic could discourage drivers to accept passengers.

## 7. Conclusions

The future of carpooling, which until now has been on a declining trend in the industrialised countries, and its capacity to decarbonise mobility practices and to foster social cohesion, are important issues for the coming years. Without claiming to exhaust the subject, this article has proposed three avenues for research. They concern a better understanding of non-work carpooling, an analysis of the many dimensions of the digitalisation of carpooling and, finally, the impact of the development of new consumption, work practices and perception of shared transport modes on the short, medium and long terms.

All these topics merit further investigation. In particular, they require the analysis of a greater variety of social and spatial contexts in a literature dominated by urban contexts, work-related carpooling and a focus on higher educated individuals. Low-density areas call for special attention, because it is here that both the environmental and the social challenges of mobility will be the greatest in the coming years for public authorities since average distances, car ownership and car use are higher than in urban settings. Practices and motivations may also be different according to education and income. Finally, to arrive at a better understanding of the different forms of carpooling, the literature needs to seek a better balance in research between carpooling for work and for other purposes, with attention to the people who practice more than one form of carpooling. Moreover, the literature should investigate the relationships between IT-based carpooling and casual carpooling (complementarity or substitution). Casual carpooling will probably continue to play a part for a few more years at least, in particular for everyday trips which rely heavily on personal networks. In any case, that is one of the many hypotheses that will need to be verified.

Beyond the academic knowledge that will result from such studies, they will also provide information for public authorities, which sometimes have a tendency to make somewhat hasty links between carpooling and the decarbonisation of travel, when research would suggest the need for greater prudence. In particular, the relationship between the different forms of carpooling, car ownership and car use need further investigation [50]. Future research should also lead to a better understanding of the social dimensions of carpooling (in terms of accessibility, social cohesion and also discriminations), which are often passed over or considered self-evident by authorities. Finally, a better consideration of the consequences of the digitalisation of carpooling and the development of new lifestyles, especially teleworking and collaborative consumption, and of the impact of the pandemic on the perception and use of shared mobility will help policymakers to implement more appropriate policies.

**Author Contributions:** Conceptualization, A.A. and E.P.; methodology, A.A.; software, A.A.; validation, A.A. and E.P.; formal analysis, A.A.; investigation, A.A.; resources, A.A. and E.P.; data curation, A.A.; writing—original draft preparation, A.A.; writing—review and editing, A.A. and E.P.; visualization, A.A. and E.P.; supervision, A.A.; project administration, A.A.; funding acquisition, A.A. All authors have read and agreed to the published version of the manuscript.

**Funding:** This research was funded by the I-site FUTURE in the framework of the Mutandis Project and the LabeX Urban Futures in the framework of the working group "Digital and the City".

**Institutional Review Board Statement:** Not applicable.

**Informed Consent Statement:** Not applicable.

**Data Availability Statement:** The data of this study is available from the authors upon request.

**Conflicts of Interest:** The authors declare no conflict of interest.

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
