# Peer review of "The Future and Sustainability of Carpooling Practices. An Identification of Research Challenges"

_sustainability, doi:10.3390/su132111824_

Round 1
Reviewer 1 Report
This article proposes a literature review on carpooling over the last decade and has been very well documented. The paper is very well prepared.
Some minor improvements are needed before the paper could be published:
- in general, the citation format is not appropriate for the journal. Please transform all citations into numeric order.
- If you write: “Indeed, there is 79 sometimes a confusion between carpooling, ridesharing and ridesourcing services in the “ (line 80) please explain, even shortly these differences.
- In my opinion the conclusion section is relatively too short. Probably the Authors could expand this part of the paper.
Author Response
This article proposes a literature review on carpooling over the last decade and has been very well documented. The paper is very well prepared.
Thank you for the compliments.
Some minor improvements are needed before the paper could be published:
- in general, the citation format is not appropriate for the journal. Please transform all citations into numeric order.
I think the editor will handle it.
- If you write: “Indeed, there is 79 sometimes a confusion between carpooling, ridesharing and ridesourcing services in the “ (line 80) please explain, even shortly these differences.
We added the following new sentences : « Indeed, in carpooling, the driver is not a professional but a private individual who agrees to share his journey (or part of it) for free or for a contribution to the travel costs. Moreover, carpooling must not be confused with carsharing, a vehicle access scheme in which people get access to a car they do not own for a period of time (Lagadic et al., 2019) ».
- In my opinion the conclusion section is relatively too short. Probably the Authors could expand this part of the paper.
We added a few sentences in the conclusion (see the manuscript with track changes).
Reviewer 2 Report
see doc uploaded

Author Response
Car pooling, or car sharing as it is more generally termed, is of interest as a potential means to reduce car ownership and use and so reduce congestion and air pollution, as the authors recognise, and importantly carbon emissions (an aspect recognised only in passing by the authors). The authors carried out a systematic literature review of social science papers, which, however, is problematic for a number of reasons: The search terms did not apparently include ‘carsharing’ and so did not identify at least one relevant paper on whether carsharing can be profitable (Lagadic et al Transport Policy 77, 68-78, 2019). More generally, use of a limited number of search terms can lead to the omission of relevant papers while at the same time generating a large number of papers that must be considered, mostly quite superficially.
It seems thatthere is a confusion between carpooling (sharing of car rides between a non professional driver and one or more passengers, for free or for a contribution to the travel costs) and carsharing. The later is a vehicle access scheme (such as Zipcar) in which people get access to a car they do not own for a period of time (one hour, a day, for instance). In order to make it clearer, we added the folllowing sentence in the introduction :
« Moreover, carpooling must not be confused with carsharing, a vehicle access scheme in which people get access to a car they do not own for a period of time (Lagadic et al., 2019) » ».
Therefore, it would not have been appropriate to include « carsharing » in the search process since it is completely different from carpooling and is therefore not the subject of this article.
Section 3, the Literature Review, is a very compressed description of the contents of the 97 selected papers without discussing the conclusions that might be drawn, which is not helpful. For instance, there is little value in stating that ‘Research has also looked at the numerous obstacles to carpooling, such as difficulties in finding a passenger or a driver and in matching schedules, but also fears or discomfort of sharing a trip with a stranger.’
The main aim of this article is to identify new research directions, and not to review the existing literature since recent meta-analysis are available. In order to make it clearer, we changed the title for « The future and sustainability of carpooling practices. A research agenda » (the word « review » does not appear anymore).
We also added the following sentence in the introduction, in order to make the goal of the paper clearer : « An in-depth analysis of this corpus highlighted the main topics addressed in current literature (section 2). They are only briefly described since the core of the paper is composed of three new avenues for research (sections 3, 4 and 5) ».
Finally, we changed the title of this section for « Brief literature review ».
The paper is unbalanced. Section 4 disproportionately comprises an extensive discussion of a single paper by the authors of the review, together with two other papers from a survey of French drivers.
This part of section 4 has been shortened and there are now less references to the two French studies that were over-represented. Here is the new paragraph : « However, few figures are available on non-work carpooling. Some studies suggest that this practice is widespread among the population, although infrequent (Gheorghiu and Delhomme, 2016; Pigalle and Aguilera, 2021). They also show that most carpoolers practice both work and non-work carpooling, something that has so far attracted very little attention in the literature, which tends to consider carpooling practices in isolation. Moreover, even if comparisons are still scarce, there seems to be a significant difference in profiles and motivations between people carpooling for work and for non-work purposes, and between short and long-distance carpooling. Therefore, these aspects merit further investigation. Based on a survey among French drivers, A. Gheorghiu and P. Delhomme (2018) highlighted that young people were over-represented among people sharing car rides for short-distance work trips, while people who shared rides for shopping purposes were in average older. In addition, women were over-represented among people who used to carpool for children-related trips. In a recent study among French adults, E. Pigalle and A. Aguiléra (2021) concluded that people carpooling for short distance non-work trips formed an older, less urban and lower earning group that people carpooling to work or for long-distance trips. In a study among Blablacar users in France (long-distance carpooling), S. Shaheen et al. (2017) demonstrated that, compared to French population, long-distance carpoolers had the same average income level but were more educated (most of the respondents had a university diploma), much younger and more frequently resided in rural municipalities. In addition, their study revealed that most shared trips were for leisure purposes. Moreover, mutual help seems to be an important motivation to share non-work related car trips, especially in areas with poor public transport provision or in communities marked by existing forms of solidarity (Blumenberg and Smart, 2014). For long-distance and short distance work-related carpooling, although there was a social component, the financial aspects seemed more influential. However, in another recent study among Blablacar users, R. Arteaga-Sanchez et al. (2020) demonstrated how social value was important to explain the continuance of use of this service”.
Section 6 is an over-long discussion of the relationship between carpooling and wide societal developments. Omission of papers concerned with algorithms to match drivers and passengers is regrettable since the effectiveness of this technology is important for the success of car sharing businesses.
As mentioned by (Adelé and Dioniso, 2020), a lot of papers about algorithms remain theoretical since they are not based on real users. They are sometimes based on intentions to use the app, which is very different. Therefore, these have not been included in our corpus.
Nonetheless, you are right that this aspect (the impacts of the nature of the algorithms and the design of the apps on real carpooling practices) is an important issue but still remains an open-question. By the way, the paper by (Adele and Dioniso, 2020), which appears in our analysis, clearly shows that some carpooling apps based on automatic matching of drivers and passengers amplify psychological barriers to carpooling. We think that more work is needed in this field, as explained at the end of section 6 : « While the literature continues to focus mainly on the technical aspects, i.e. improvements in the matching algorithms and in the design of the apps, there is a shortage of more qualitative studies that take a social perspective on the obstacles to IT-based carpooling for work and non-work purposes (Adelé and Dionisio, 2020) ».
Exclusion of non-academic material means that factors relevant to the success of car sharing are not fully considered. For instance, a recent UK survey found that 80% of respondents expressed concern at sharing a car journey or shared use of a car https://www.gov.uk/government/publications/future-of-transport-deliberative-research The current interest in carpooling/sharing is mainly concerned with the possibility that this may reduce individual car ownership and use, thus contributing to a reduction in transport carbon emissions.
The exclusion of non-academic material is a methodological choice. In addition, the paper is not on the factors relevant to the sucess of carpooling (and not carsharing), but more on some future research challenges on its future trends and impacts. I hope we made it clearer in the introduction and by (slightly) changing the title.
At the beginning of section 7 the authors recognise that carpooling has been on a declining trend. A useful review would attempt to explain this trend and identify actions that might reverse it, whether new business models or research that would illuminate the barriers and opportunities. Instead, the authors treat the topic as primarily one for academic study, which limits the likely readership.
As explained in section 3 (Brief literature review), the barriers to carpooling is the topic of a considarable amount of papers (and literature reviews). Our purpose is different. We want to highlight important research challenges for the future on carpooling practices.
Reviewer 3 Report
This study is aimed to propose a literature review on carpooling over the last decade. This article needs some work for readers to have a better understanding. The authors may consider the following comments to improve the quality of paper.
- Since carpooling may be culture-oriented, it is recommended to review this part in the article.
- The review of carpooling in the "health crisis" issue is relatively shallow, and more materials in depth can be added.
- The research proposes links to collaborative consumption, telecommuting, and health crises. However, a more convincing explanation and whether there are other issues related to carpooling can be reinforced.
- It is recommended that the references cited in the article need to be listed completely.
- For carpooling in the future research, the authors need to propose more specific and practical directions and content.
- Please add one reference: Chen, T.-Y.; Jou, R.-C.; Chiu, Y.-C. Using the Multilevel Random Effect Model to Analyze the Behavior of Carpool Users in Different Cities. Sustainability2021, 13, 937. https://doi.org/10.3390/su13020937.
Author Response
This has been done.
Round 2
Reviewer 2 Report
The authors have clarified the intention of the paper, which is not a review of the literature aimed at explaining why carpooling has been on a declining trend, nor does it consider how car pooling could contribute to reducing transport carbon emissions, a topic of policy importance.
The paper corresponds more to a grant proposal to justify three avenues of research. Were that research to be carried out, the material in the paper would provide context in which to present the research findings. In the absence of such findings, the interest in the paper is likely to be limited.
Author Response
The authors have clarified the intention of the paper, which is not a review of the literature aimed at explaining why carpooling has been on a declining trend, nor does it consider how car pooling could contribute to reducing transport carbon emissions, a topic of policy importance.
Response: We changed the title again, in order to make it clearer : « The future and sustainability of carpooling practices. An identification of research challenges ». We also modified the abstract accordingly.
The paper corresponds more to a grant proposal to justify three avenues of research. Were that research to be carried out, the material in the paper would provide context in which to present the research findings. In the absence of such findings, the interest in the paper is likely to be limited.
Response: This type of paper, which objective is to identify new research challenges, is common in academic publications, and in general very useful for researchers (and especially PhD students). This is not a grant proposal, but more a contribution to what topics we should consider in the future.
See for instance the following references, in the transportation field :
Hussain, E., Bhaskar, A., & Chung, E. (2021). Transit OD matrix estimation using smartcard data: Recent developments and future research challenges. Transportation Research Part C: Emerging Technologies, 125, 103044.
May, A. D., Jopson, A. F., & Matthews, B. (2003). Research challenges in urban transport policy. Transport Policy, 10(3), 157-164.
van Wee, B., & Ettema, D. (2016). Travel behaviour and health: A conceptual model and research agenda. Journal of Transport & Health, 3(3), 240-248.
Reviewer 3 Report
Please provide authors' responses to the reviewer's comments points by points.
Author Response
This study is aimed to propose a literature review on carpooling over the last decade. This article needs some work for readers to have a better understanding. The authors may consider the following comments to improve the quality of paper.
Response: Thank you very muh for your valuable comments on our paper.
- Since carpooling may be culture-oriented, it is recommended to review this part in the article.
Response: Thank you very much for this suggestion. A sentence with two references were added in the new version : « Cultural elements, such as car culture (Madubuike, 2017), play also a role in explaining carpooling behavior, but they were far less investigated (Pinto et al., 2019) ».
- The review of carpooling in the "health crisis" issue is relatively shallow, and more materials in depth can be added.
Response: Following your recommendation, we decided to integrate the telework issues into the health crisis issue, which is now structured into two topics : teleworking and modal choice. Moreover, we added material to the section about modal choice. Here is the new version of the text regarding « modal choice issues after the pandemic » :
Travel impacts of the pandemic remain highly uncertain and need further investigation regarding the demand for private versus shared transportation modes (Shokhouyar et al., 2021). Nonetheless, shared modes are threatened since they represent a potential risk of contamination. In that respect, public transport has received much attention than other shared transport modes. Huge ridership reductions were observed many countries across the world, during but also after lockdown periods (Gkiotsalitis and Cats, 2021). Studies are needed about the impacts on the other shared modes such as carpooling. In particular, carpool could be less impacted than public transport or even benefit from the situation since it could be considered as less risky than public transport (Conway et al. 2020; Molina et al., 2020; Tomas et al., 2021) due to fewer people or because carpooling apps could require sanitary guarantees such as health pass. On the other hand, carpool could be considered as a risky transport mode compared to solo driving. In particular, the pandemic could discourage drivers to accept passengers.
- The research proposes links to collaborative consumption, telecommuting, and health crises. However, a more convincing explanation and whether there are other issues related to carpooling can be reinforced.
Response: The new version the introduction now includes the following explanation about the choice of the research challenges proposed in this article :
« Without pretention of exhaustiveness, this article proposes several new research directions regarding the future and sustainability of carpooling practices. The reflection is based, on the one hand, on a systematic review of literature (2010-2021) in social sciences, using Google Scholar and a list of relevant keywords, and, on the other hand, on a consideration of some of the main recent changes in carpooling itself, and in lifestyles that may affect carpooling practices. On the one hand, carpooling practices are affected by digitalization: the rise of carpooling apps and platforms and their progressive integration into mobility platforms, such as MaaS (for Mobility-as-a-Service) platforms, but also the use social media by carpoolers. On the other hand, we think that carpooling practices should be affected by the rise of collaborative consumption in general, and by the consequences of current health crisis on telework and modal choice. »
The abstract was also modified in order to better explain our approach.
- It is recommended that the references cited in the article need to be listed completely.
Response: The references were checked.
- For carpooling in the future research, the authors need to propose more specific and practical directions and content.
Each section is now better structured around precise research questions, especially regarding non-wrok carpooling and the digitalisation of carpooling. We also deleted some sentences that were to general. Finally, we made the links with health crisis more clearer by integrating the teleworking part into this topic.
- Please add one reference: Chen, T.-Y.; Jou, R.-C.; Chiu, Y.-C. Using the Multilevel Random Effect Model to Analyze the Behavior of Carpool Users in Different Cities. Sustainability2021, 13, 937. https://doi.org/10.3390/su13020937.
Response: The reference was already cited in the former version of the text (Section 3 – Brief literature review). We also added it in the conclusion of the new version.
Round 3
Reviewer 3 Report
The comments raised by the reviewer have been responded properly.